# LEARNING TO IMPROVE CODE EFFICIENCY

## ABSTRACT

Improvements in the performance of computing systems, driven by Moore's Law, have transformed society. As such hardware-driven gains slow down, it becomes even more important for software developers to focus on performance and efficiency during development. While several studies have demonstrated the potential from such improved code efficiency (e.g., 2x better generational improvements compared to hardware), unlocking these gains in practice has been challenging. Reasoning about algorithmic complexity and the interaction of coding patterns on hardware can be challenging for the average programmer, especially when combined with pragmatic constraints around development velocity and multi-person development.

This paper seeks to address this problem. We analyze a large competitive programming dataset from the Google Code Jam competition (Google Code-Jam) and find that efficient code is indeed rare, with a 2x runtime difference between the median and the 90th percentile of solutions. We propose using machine learning to automatically provide prescriptive feedback in the form of hints, to guide programmers towards writing high-performance code. To automatically learn these hints from the dataset, we propose a novel discrete variational auto-encoder, where each discrete latent variable represents a different learned category of code-edit that increases performance. We show that this method represents the multi-modal space of code efficiency edits better than a sequence-to-sequence baseline and generates a distribution of more efficient solutions.

## 1 INTRODUCTION

The computational efficiency of code is often front-and-center in any computer science curriculum. While there are many ways to solve a particular problem, there is often wide variance in the runtime of different implementations. This variance is often attributed to many different factors: the algorithmic complexity of the code in question, the data structures that are used, the libraries that are called, and lower-level execution effects like efficient caching or memory usage.

Similarly, computational efficiency is a critical component of professional software development. The computing industry as a whole has relied on the automatic performance increases of Moore's Law to scale massive warehouse computing systems to meet the internet requirements of the world. As these automatic performance increases slow down, the burden of reducing computational cost and carbon footprint now falls on writing high-performance code (Patterson et al. (2021)).

Writing efficient code is challenging, even for experienced programmers, as it requires understanding computational complexity as well as the underlying hardware. Lower-level performance optimizations are therefore automated by compilers which automatically apply a small set of known, sound low-level program transformations to an already written program to increase its efficiency. However, compilers and current tooling have more difficulty identifying higher-level optimizations, such as more efficient algorithms for the same problem. So far, these types of optimizations could only be identified by humans. We hypothesize that machine learning can be used to guide humans towards such optimizations, by suggesting edits that optimize code efficiency.

To study this problem, we examine a competitive programming dataset where tens of thousands of developers have submitted answers to about 180 different questions. Studying these solutions, we find wide variance in computational cost: the runtime difference between a median solution and the 90th percentile is over two-fold. The scarcity of high-performance solutions highlights the difficulty of our task. Therefore, we aim to provide prescriptive feedback to developers to guide them towards writing high-performance code.

We develop a framework to apply multiple categorical transformations to a single program using a novel discrete variational autoencoder where different vectors in the latent dictionary lead to different code transformations. We find that these learned categories are often consistent (e.g., a particular latent variable may control the data structure that is used for a particular problem or a for-loop vs. a while-loop), and that by applying these transformations to the program, we can move solutions into a more efficient computational efficiency category vs. the code that the developer wrote.

This paper makes the following contributions:

- We frame code efficiency optimization as a generative problem.
- Using the Google Code Jam competitive programming dataset (Google Code-Jam), we analyze the distribution and characteristics of high-performance solutions. We find that high-performance solutions are uncommon and consist of a combination of many distinct optimizations. We then derive a canonicalized program-edits dataset to train models to improve code efficiency.
- We propose a novel discrete generative latent-variable model of program edits to model a distribution of fast programs, conditioned on their slower counterparts. We find that this model outperforms a sequence-to-sequence baseline along three different axes: correctness, efficiency, and diversity.
- We qualitatively demonstrate that the learned discrete latent variables represent different edits, and that the edits that are assigned to one latent variable are generally consistent. As a side-effect, we learn an interpretable program embedding space.

We believe that these results are a promising step towards automating the process of identifying and applying higher-level performance optimizations, which would fundamentally increase the capabilities of current developer tools while reducing the carbon footprint of computing.

## 2 BACKGROUND

### 2.1 PROBLEM FORMULATION

There are many different ways to implement a particular algorithm. For algorithms like matrix multiplication, small syntactic changes like loop reordering have a dramatic impact on execution cost (Leiserson et al. (2020)). From low-level hardware effects like caching and branch prediction, to higher-level code choices like data structures, termination conditions, and loops - navigating the space of implementation options is a key element of software engineering.

We find many of these performance archetypes when looking at competitive programming solutions. Figure 1 shows three example programs on the left and faster versions of those programs on the right. The first example showcases using a more efficient datastructure (a heap), which then enables early termination of the main loop. The second example highlights a performance bug, where fewer API calls can accomplish the same task. The third example highlights how using built-in libraries can be faster than writing bespoke implementations. These examples represent just three of the many discrete design choices that developers make while coding their solutions. We hypothesize that these discrete choices can be learned, such that a generative model can suggest different code transformations that a developer could leverage to increase code efficiency.

### 2.2 PROGRAM EDIT DATASET

To study this problem, we use the dataset from the Google Code Jam international competitive programming competition (Google Code-Jam). Each question consists of a problem description, along with up to three test cases – inputs and desired outputs – of increasing complexity. For each question, the dataset contains solutions from competition participants. If the submission passes each test, it is labeled as correct and annotated with run-time, otherwise it is marked as incorrect.

For our study, we focus on the solutions that are written in Python. As we aim to study execution complexity, we focus only on correct submissions and consider the run-time of the largest test case. This distribution of run-times is shown in Figure 2.

Even for this constrained competitive programming task (with a natural focus on efficient solutions), Figure 2(a) illustrates a wide distribution in run-time – supporting our hypothesis that writing efficient

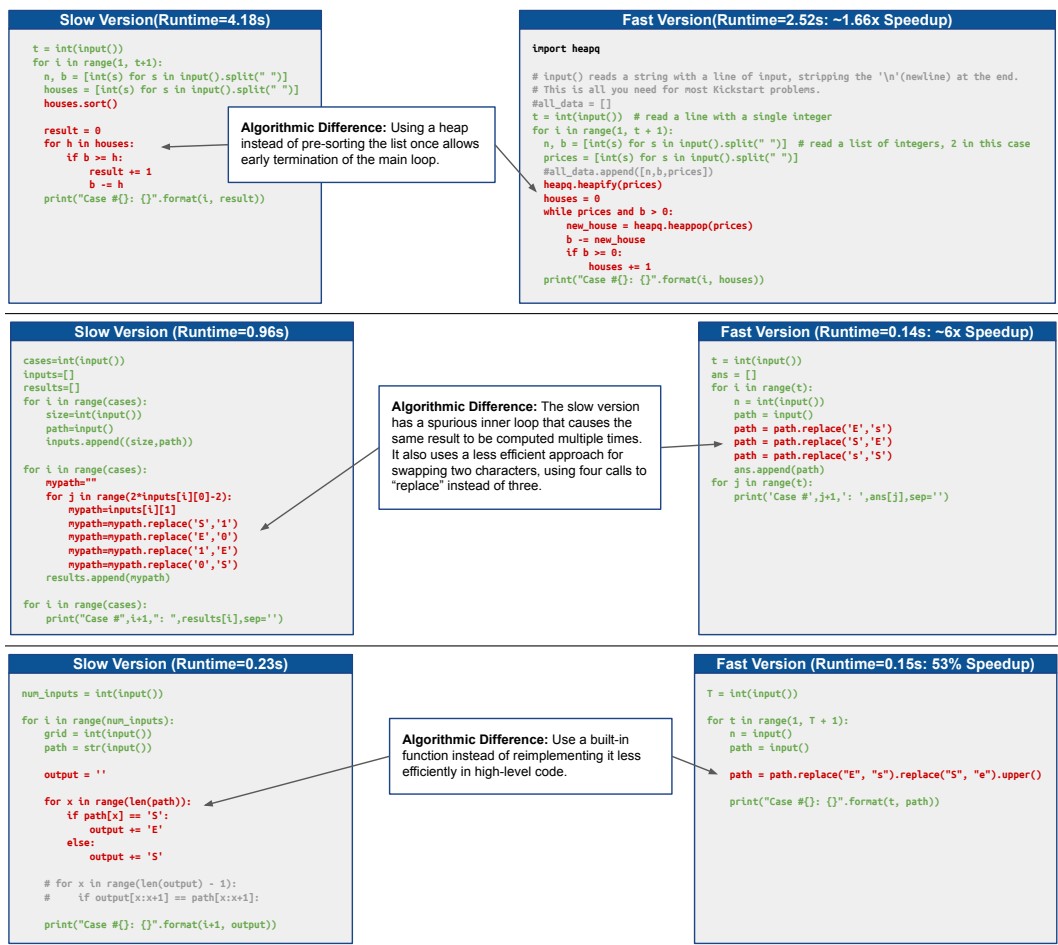

Figure 1: Examples from the program pair dataset. Each row corresponds to a data point, containing the input (slow) submission and the output (fast) submission.

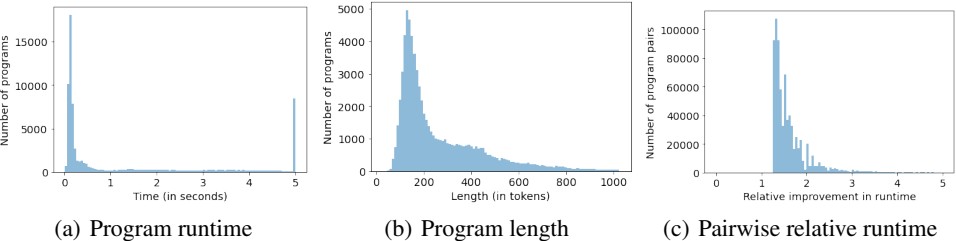

(a) Program runtime    (b) Program length    (c) Pairwise relative runtime

Figure 2: Dataset statistics.

code is a difficult problem. We observe solutions with run-times that are fractions of a second, as well as those that time-out after 5 seconds. High-performance solutions are rare. Additionally, Figure 2(b) indicates that the data is multi-modal with a wide textual distribution of solutions. Solutions are as short as a handful of lines of code and stretch to over a thousand. We aim to discover the common patterns in these solutions that lead to efficient code, so that we can provide hints to programmers to improve performance.

We frame this task as a sequence-to-sequence problem: given an input code sequence, output a more efficient version of the code. While sequence-to-sequence modeling is often posed as a one-to-one problem, our task is a one-to-many problem, as there are an overwhelmingly large number of such solutions to a single problem. Trivially, one could simply rename variables or change syntax. We

would like to focus on substantive changes, so we deal with this in two ways: first, we canonicalize the code submissions by renaming variables, function names, and strings with generic tokens. Next, we construct a dataset of inputs and outputs by pairing programs according to their similarity. Specifically, we use the ROUGE-8 metric to score the similarity between all pairs of canonicalized submissions. Similar programs with a significant run-time difference (we choose 1.2x speedup or larger) are added to the dataset. Figure 2(c) shows the relative runtime difference between the slow/fast pairs in the dataset. Consistent with 2(a), we observe a broad distribution of run-time differentials.

We find that this canonicalization approach enables the model to learn on this dataset, but also means that we lose the ability to execute the edited program to evaluate run-time. However, even in canonical form, edits are qualitatively sufficient to identify optimizations. Section 4.3 describes the quantitative metrics that we use to further evaluate the quality of these edits.

## 3 METHOD

### 3.1 MOTIVATION

Our goal is to create a model that can condition on a given program and help developers identify more computationally efficient variants of that program. There are three factors that we use to drive our modeling choices and evaluation. 1) The resultant hints should be syntactically coherent. As the model provides suggestions, the suggested changes do not need to be perfect, but still need to make sense. 2) The changes suggested by the model should lead to more computationally efficient code. To quantify efficiency, we rely on textual similarity to other known correct code in the data-set that is more efficient. 3) For a given source program, there are a multitude of discrete transformations that could lead to more efficient code (that is, the problem is one-to-many). For example, more efficient code versions could change the data structures involved, wrap/unwrap code in functions, or adjust loop bounds. These edits are not known a priori, so the model should automatically learn which transformations increase efficiency, and then communicate those choices to developers. Therefore, an ideal model would be able to generate many different distinct hints, so we consider code diversity as an auxiliary objective. The criteria mentioned above (efficiency, correctness, diversity) inform our modeling choices. In Section 4.3, we describe specific metrics to quantify each of these criteria.

### 3.2 NOTATION

We denote an input sequence $\mathbf{x} = [x_1, x_2, ..., x_{T_{in}}]$ and an output sequence $\mathbf{y} = [y_1, y_2, ..., y_{T_{out}}]$, where $x_i, y_j$ are tokens in a vocabulary $\mathcal{V}$. We will represent the encoder as $f_\theta(\cdot)$ and the decoder as $g_\phi(\cdot)$, where $\theta$ and $\phi$ are the parameters of the networks. As is standard in sequence modeling, we embed tokens using an embedding function $e(\cdot)$ and add positional encodings $PE$. Unless it is informative to say so, we will assume that this is done implicitly.

### 3.3 BASELINE MODEL: TRANSFORMER

Prior work in program optimization often operates at the compiler or assembly-code level (Schkufza et al., 2013). As we aim to provide much higher-level natural language hints to edit source code instead, we use a state-of-the-art natural language model as a baseline (Vaswani et al., 2017). The baseline Transformer is trained in a sequence-to-sequence fashion, where given a pair of programs with runtimes $r_{slow}$ and $r_{fast}$, we use the slower program as the input $\mathbf{x}$, and the faster program as the output $\mathbf{y}$. We encode the input program using $f_\theta(\mathbf{x})$ and decode the output program using $g_\phi(f_\theta(\mathbf{x}))$. We train the network to minimize cross entropy loss. To generate different programs given one input program, we simply randomly sample outputs.

### 3.4 DISCRETE VARIATIONAL AUTO-ENCODERS

As an alternative to a fully black-box solution, we propose solving the problem with a more explicit and interpretable model. Since we observe in the dataset that there are many discrete categories of efficiency-improving code transformations, we propose using a discrete variational auto-encoder to learn these categories in an unsupervised way.
Furthermore, we are interested in learning fine-grained *edits* of code, where each category of learned edit ideally represents a single conceptual or localized change. Towards that end, we encode $\mathbf{x}$

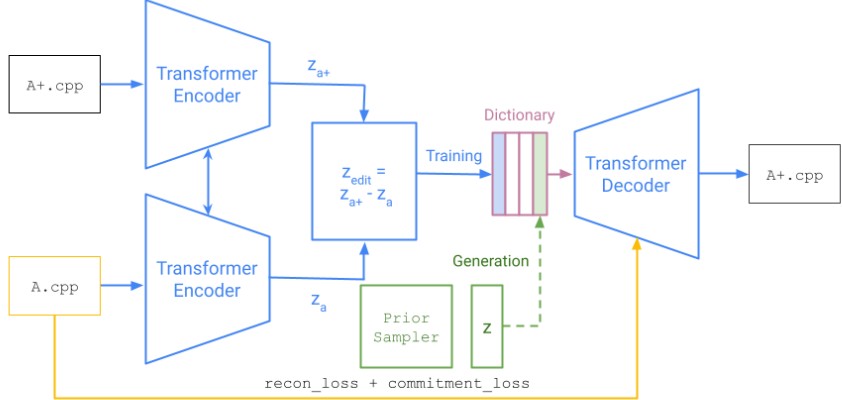

Figure 3: Conditional program edit VQ-VAE.

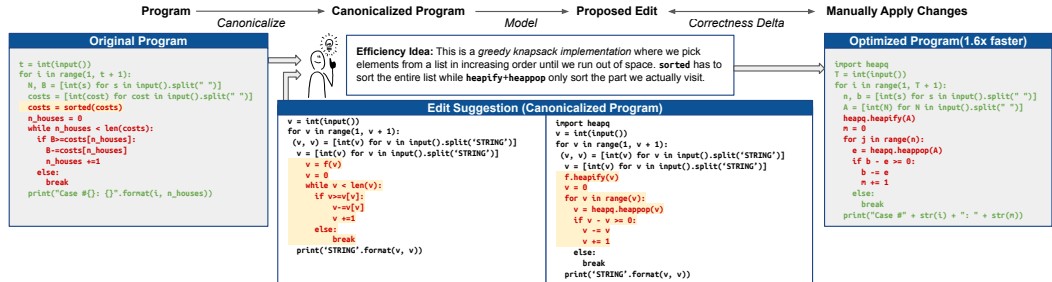

Figure 4: An end-to-end example of the model's output and its intended use by a programmer. The model *directionally* suggests a code efficiency optimization for the programmer to consider.

and $\mathbf{y}$ into latent vectors $\mathbf{z}_{\text{fast}}$ and $\mathbf{z}_{\text{slow}}$, and take their difference $\mathbf{z}_{\text{edit}} = \mathbf{z}_{\text{fast}} - \mathbf{z}_{\text{slow}}$ to capture the necessary information to edit $\mathbf{x}$ into $\mathbf{y}$. This two-tower encoding approach with tied encoders allows the model to learn a relative edit encoding, which we find helps training stability. We find that code canonicalization aids in learning this relative encoding, as it removes semantic differences between the input and output programs. The embedding vectors $\mathbf{z}_{\text{fast}}$ and $\mathbf{z}_{\text{slow}}$ are computed using a multi-headed attention layer between the encoded outputs from $f_\theta$ and a learned query vector $c$. This form of pooling has been found in other works as well (Lee et al., 2019).

To capture the one-to-many relationship of code edits, and to further emphasize the idea of broad and interpretable edit categories, we choose a VQVAE to quantize $\mathbf{z}_{\text{edit}}$ (Van Den Oord et al., 2017). This allows us to train a generative model of $K$ discrete edit types. Each type, when applied to a piece of code, produces a different result. This also has the advantage of preventing posterior collapse, a pervasive issue with variational auto-encoders, especially with respect to discrete sequence modeling.

Specifically, we learn a dictionary of embedding vectors $Z = [\mathbf{z}_1, \mathbf{z}_2, \dots, \mathbf{z}_K]$. When encoding the difference vector, we set $\mathbf{z}_{\text{edit}} = \arg\min_k \|\mathbf{z}_k - (f_\theta(\mathbf{y}) - f_\theta(\mathbf{x}))\|_2$. Our Edit-VQVAE model is illustrated in Figure 3.4 and our forward and decoding procedure is outlined in Algorithms 1 and 2 in the Appendix. To support this model, we modify the decoder to take $\mathbf{z}_{\text{edit}}$ as an auxiliary input. The total input to the decoder is therefore the sum of the token embeddings, positional encodings, and difference vector: $e(\mathbf{x}) + PE + \mathbf{z}_{\text{edit}}$.

## 4 EVALUATION

### 4.1 METHODOLOGY

The primary utility of our model is to provide hints to programmers that identify improvement opportunities for their code. This means that changes suggested by the model do not need to be

perfect; they just need to point programmers in the right direction. We evaluate these hints first qualitatively in Section 4.2 then quantitatively in Section 4.3.

**Experimental setting.** The program pair dataset for training and evaluating program optimization tasks is constructed from the raw dataset following the procedures described in Section 2.2. After processing the raw program submission dataset into program pairs, we randomly split the program pairs into training, validation and testing set. The training/validation sets contain $806,041/1,458$ program pairs. The test set contains 100 input programs that are not visible during training. The program pair runtime distribution is illustrated in Figure 2(c).

We implemented all models in Jax (Bradbury et al., 2018), and trained the models using a peak learning rate of 0.01 with both warmup and decay schedules. All models were trained with a batch size of 16 for 100 epochs, using distributed data parallel training on 64 Google Cloud TPU cores and 16 host machines. We performed rudimentary hyperparameter tuning for all models. For all Transformer blocks, we set the dropout rate to $p_{\text{dropout}} = 0.1$, the attention layer embedding size to $d_{\text{model}} = 128$, and the feed-forward inner layer size to $d_{\text{ff}} = 512$. We use 6 Transformer layers, each with 8 attention heads, for both encoders and decoders. The Edit-VQVAE has $K = 64$ latents. In the quantitative evaluation, we randomly sample $N = 64$ programs from each model for evaluation. The Edit-VQVAE generates 1 program per latent.

## 4.2    EDIT VQ-VAE QUALITATIVE ANALYSIS

One of the main benefits of a discrete latent variable model is an interpretable latent space that we can use to visualize the learned efficiency archetypes. After training and during generation, we remove the encoder and directly feed the canonicalized program into the decoder along with a categorical variable that selects one of 64 discrete latent edits. A real learned example of this is shown in Figure 4. The canonicalized program is edited by the decoder to increase performance (in this case the model suggests a heap). We find that these edits expose the key optimization idea that can then be translated back into code by the programmer.

We often find that each of the discrete latents is responsible for a single syntactic change (adding a heap data structure, changing a for-loop to a while-loop and vice versa, adding functions, etc.) – and that we can take different transformations on the same program by varying the latent code provided to the VQ-VAE. The model learns many different efficiency archetypes (Table 2), including those discussed in Section 2.

We can visualize the effect that the transformations have on different program pairs through PCA decomposition as well (shown in the Appendix). If the latent variable has learned to apply a similar edit to its input programs, one would expect that the slow programs would be collectively moved to a faster region of latent space. We observe this behavior in Figure 5.

## 4.3    QUANTITATIVE RESULTS

Beyond the qualitative analysis in Section 4.2, we quantitatively evaluate the suggestions provided by the Edit VQ-VAE vs. a vanilla sequence-to-sequence Transformer on three axes: correctness, efficiency, and diversity.

**Evaluation metrics.** We use syntactic program similarity as a tool to measure model performance. Program similarity is computed between the generated programs and a set of reference programs, where the reference programs are programs in the dataset that solve the same question more efficiently than the original input program. For each input program $x$ in the test set, we construct its reference program set as follows,

$$\mathcal{R}(x) = \{y \in \mathcal{D}_{q_x} \mid \text{runtime}(y) < \text{runtime(x) and ROUGE\_8}(y, x) < t\}, \tag{1}$$

where $\mathcal{D}_{q_x} \in \mathcal{D}$ is the subset of program submissions in the dataset $\mathcal{D}$ that solves the same question $q_x$ as input program $x$, and $t > 0$ is a threshold for the program neighborhood. We propose the axis-specific metrics below, which are built upon conventional text similarity metrics such as the BLEU score (Papineni et al., 2002) and the $\Delta$BLEU score (Galley et al., 2015).

- Correctness Delta, *i.e.*, similarity between the generated program and an existing solution to the same question as the input program. We adopt the BLEU score here as a surrogate. For any

| | Average results (64 Samples) | | | Maximum results | | | |
|---|---|---|---|---|---|---|---|
| | Corr. | Effi.-hard | Effi.-soft | Corr. | Effi.-hard | Effi.-soft | Diversity |
| Edit-VQVAE | **0.760** | **0.059** | **0.765** | **0.867** | **0.359** | **0.875** | **0.807** |
| VQVAE | - | - | - | - | - | - | - |
| Edit-VAE | - | - | - | 0.785 | 0.027 | 0.789 | - |
| Transformer | 0.572 | 0.006 | 0.608 | 0.733 | 0.125 | 0.769 | 0.655 |
| Transformer* | 0.575 | 0.015 | 0.611 | 0.769 | 0.242 | 0.808 | 0.684 |

Table 1: Performance of the proposed model and baselines measured by correctness (Corr.), efficiency (Effi.), and diversity metrics. Numbers are between 0 and 1, higher is better. The best result is bolded. The numbers from the first three columns are obtained by taking the average over the metric scores of all $N$ samples, the middle three columns by taking the maximum. The last column is computed based on all $N$ samples. The VQVAE does not converge and thus results in poor performance (denoted by a '-'); Edit-VAE suffers from severe posterior collapse, so there is only one sample generated from the model. Hard metrics require exact matches while soft metrics use $\Delta$BLEU.

generated program $\hat{y}$, the correctness score is defined as,

$$S_C(\hat{y}, x) \triangleq \text{BLEU}(\text{candidate} = \hat{y}, \text{references} = \mathcal{R}(x)). \tag{2}$$

$S_C$ gives a score in $[0, 1]$, a score of 1 indicates there is a perfect overlap between $\hat{y}$ and the reference programs in $\mathcal{R}(x)$, while a score of 0 means there is no overlap. A higher $S_c$ score indicates that the generated program is more similar to the reference programs, implying less programmer effort to translate the suggestion into a correct program.

- Efficiency Delta, *i.e.*, similarity between the generated program and an existing more efficient solution to the same question as the input program. There are two considerations: 1) how much more efficient an alternative program is compared to the input, and 2) how different this alternative program is to the suggestion. We adopt two metrics for efficiency, a hard metric using an exact match, and a soft metric using $\Delta$BLEU.

  - If the generated program $\hat{y}$ has an exact match with one of the reference programs in $\mathcal{R}(x)$, we can directly use its runtime to compute the efficiency for $\hat{y}$. The exact match score is defined as,

  $$S_{EH}(\hat{y}, x) \triangleq \max_{y \in \mathcal{R}(x)} \mathbb{1}[\hat{y} == y] \frac{\min_{y' \in \mathcal{R}(x)} \text{runtime}(y')}{\text{runtime}(y)}. \tag{3}$$

  $S_{EH}$ gives a score in $[0, 1]$, a score of 1 indicates an exact match with the most efficient program in the reference set, while a score of 0 means no exact match.

  - Requiring generated programs to be exactly the same as a reference program will undervalue good suggestions that are close to an efficient solution, but not exactly the same. Therefore we also employ the $\Delta$BLEU score as a soft version for efficiency. $\Delta$BLEU is a variant of BLEU where each reference can be assigned with a unique weight indicating the quality of the reference. The score is defined as,

  $$S_{ES}(\hat{y}, x) \triangleq \Delta\text{BLEU}(\hat{y}, \mathcal{R}(x), \text{weight}(y) = \frac{\min_{y' \in \mathcal{R}(x)} \text{runtime}(y')}{\text{runtime}(y)}). \tag{4}$$

  A reference program is assigned a higher weight if it is more efficient. $S_{ES}$ is in $[0, 1]$, a score of 1 indicates a perfect overlap between $\hat{y}$ and a more efficient reference program. As with correctness, this approximates how much additional work would be required to transform the suggestion into an efficient solution.

- Diversity, *i.e.*, whether the model can generate diverse efficient programs given one input program. We define diversity as the model's ability to recover the set of reference programs using the generated programs $\hat{\mathcal{Y}}(x) = \{\hat{y_1}, \hat{y_2}, \cdots, \hat{y_n}\}$. The diversity score is defined as,

$$S_D \triangleq \frac{1}{|\mathcal{R}(x)|} \sum_{y \in \mathcal{R}(x)} \max_{\hat{y} \in \hat{\mathcal{Y}}(x)} \text{BLEU}(\hat{y}, y). \tag{5}$$

$S_D$ is in $[0, 1]$. A score of 1 indicates perfect recovering of the reference set.

Our results are summarized in Table 1. For metrics on correctness and efficiency, we show both average and maximum performance over the $N$ samples. We find that Edit-VQVAE outperforms the sequence-to-sequence Transformer baseline by a large margin on all metrics. As a one-to-many model, the Edit-VQVAE is able to learn the program edit distribution from program pairs that share the same input programs in the training dataset. Conversely, sequence-to-sequence models perform better when learning a one-to-one mapping with a single mode, but do not guarantee the quality of multiple random samples. This is supported by our data showing that the gap between the Transformer baseline and the Edit-VQVAE is smaller on the maximum results than on the average results.

The Transformer baseline is trained using all the program pairs in the same training set as the Edit-VQVAE. What if the Transformer baseline is only trained on the program pairs with the largest relative performance? Would its maximum results outperform Edit-VQVAE? Interestingly, the answer is no. For program pairs that share the same input program in the training set, we only keep the pair with maximum runtime improvement. This filtered dataset is used to train another Transformer model, denoted as Transformer*. Transformer* significantly outperforms Transformer on the maximum metrics, but it still underperforms Edit-VQVAE.

To understand the effect of the design decisions in Edit-VQVAE, we perform an ablation study by removing the discrete and edit-based structure in the model. This results in two ablated models: 1) VQVAE without the edit structure, and 2) Edit-VAE. In the VQVAE, the latent is obtained by directly feeding the concatenation of the program pairs into the Transformer encoder, leading to a model with $4x$ parameters. In the Edit-VAE, the discrete latent space is replaced by the original continuous Gaussian latent space. Empirically, we found that it is difficult for VQVAE to converge. We hypothesize this is due to the fact that it is hard to control the scale of the output of a Transformer encoder block. The edit-based structure instead uses a relative encoding, which potentially controls the scale of the encoder output, and helps stabilize training. Additionally, we found that the Edit-VAE model suffers from posterior collapse and is not able to generate more than one program during test time. Edit-VQVAE is more robust to posterior collapse due to a discrete latent space (Van Den Oord et al., 2017).

## 5 RELATED WORK

**Program Synthesis** Program synthesis involves automatically writing code from a program specification. Many deep learning approaches have been proposed in recent years (Balog et al., 2016; Bunel et al., 2018; Devlin et al., 2017b; Kalyan et al., 2018; Devlin et al., 2017a; Lee et al., 2018; Nye et al., 2019; Odena & Sutton, 2020; Parisotto et al., 2016). However, these works generally focused on writing code to satisfy a given specification without regard for its efficiency. There are neural (Zhao et al., 2018) and non-neural (Meng et al., 2011) models that edit code, but these are used for different applications such as fixing errors (Yasunaga & Liang, 2020; Chen et al., 2021b).

**Language models and large-scale program synthesis** Large language models (LLMs) built on Transformers (Vaswani et al., 2017) and trained with massive amounts of data have begun to yield significant improvements in many natural language tasks (Chowdhery et al., 2022). These breakthroughs have been ported over to program synthesis where LLMs have demonstrated an impressive ability to solve coding challenge problems (Austin et al., 2021; Chen et al., 2021a). These approaches have been concerned with generating a program from a high-level specification, as opposed to editing an existing program to make it faster.

**Other forms of optimization** Program optimization has been applied to efficiency-related areas: Query optimization (e.g., using deep learning (Krishnan et al., 2018; Marcus et al., 2019)) seeks to optimize query execution plans for database systems. Stochastic superoptimization (Schkufza et al., 2013) uses random search to optimize x86 assembly code. Assembly and queries are both highly structured languages and the programs are loop-free and quite small. We aim to optimize higher-level source code, which translates to vastly larger assembly programs, using a generative approach.

**Discrete representations** Many real-world problems are discrete, and discrete representations can often be more composable and interpretable. Learning discrete representations using generative modeling is challenging due to the lack of gradients, necessitating the use of gradient estimators like the Gumbel-softmax (Jang et al., 2016; Maddison et al., 2016) and REINFORCE (Williams, 1992). The VQ-VAE (Van Den Oord et al., 2017) avoids this by using vector quantization and treating

(a) A transformation that adds an early termination to a while loop. We see that the model also applies other semantics-preserving transformations that can be safely ignored.

| Original | Edit #1 | Edit #2 |
|---|---|---|

```
for v in range(int(input())) :
    (v , v) = [ int(v) for v in input() . split() ]
    v = list(map(int , input() . split()))
    v = 0
    v.sort()
    for v in v :
        if(v <= v) :
            v -= v
            v += 1
    f('STRING'. format((v + 1) , v))
```

```
for v in range(int(input())) :
    (v , v) = [ int(v) for v in input() . split() ]
    v = list(map(int , input() . split()))
    v.sort()
    v = 0
    for v in v :
        if(v <= v) :
            v -= v
            v += 1
        else:
            break
    f(f 'STRING{(v + 1)}STRING{v}')
```

```
v = int(input())
for v in range(v) :
    (v , v) = [ int(v) for v in input() . split() ]
    v = list(map(int , input() . split()))
    v = 0
    v.sort()
    for v in v :
        if(v <= v) :
            v -= v
            v += 1
        else:
            break
    f('STRING' . format((v + 1) , v) , flush = True)
```

(b) A notable change this transformation identified was to replace a `[ ]` operator with a custom lambda using a `map` operation with a builtin, which is indeed a pattern that should lead to speed-ups.

| Original | Edit #1 | Edit #2 |
|---|---|---|

```
for v in range(int(input())):
    (v, v) = [int(v) for v in input().split()]
    v = list(map(int, input().split()))
    v = 0
    v.sort()
    for v in v:
        if(v <= v):
            v -= v
            v += 1
    f('STRING'.format((v+1), v))
```

```
for v in range(int(input())):
    (v, v) = map(int, input().split())
    v = list(map(int, input().split()))
    v.sort()
    v = 0
    for v in v:
        if(v <= v):
            v -= v
            v += 1
        else:
            break
    f(f'STRING{(v+1)}STRING{v}')
```

```
v = int(input())
v = 0
while (v > 0):
    v += 1
    (v, v) = [int(v) for v in input().split()]
    v = list(map(int, input().split()))
    v = 0
    v.sort()
    for v in v:
        if(v <= v):
            v -= v
            v += 1
    f('STRING'.format(v, v))
    v -= 1
```

(c) In the first sample, the model identified an optimization to expressing a loop condition. In the second example, it identified a different way to express integer division (note that it even added the required `math` import).

| Original | Edit #1 | Edit #2 |
|---|---|---|

```
def f() :
    for v in range(1, (int(input()) + 1)):
        (v, v) = [int(v) for v in input().split()]
        v = [int(v) for v in input().split()]
        v = v
        for v in f(range(len(v))) :
            v = ((v // v[v]) * v[v])
        f(((('STRING' + f(v)) + 'STRING') + f(v)))
    if(f == 'STRING') :
        f()
```

```
def f(v, v):
    v = (len(v) - 1)
    while (v > 0):
        v =((v // v [ v ]) * v [ v ])
        return v
def f():
    for v in range(1, (int(input()) + 1)):
        (v, v) = [int(v) for v in input().split()]
        v = [int(v) for v in input().split()]
        v = f(v , v)
        f(((('STRING' + f(v)) + 'STRING') + f(v)))
    if(f == 'STRING'):
        f()
```

```
import math
def f(v, v, v):
    v = v
    for v in f(range(v)):
        v = (f.floor((v / v [ v ])) * v [ v ])
        return v
v = int(input())
for v in range(1, (v + 1)):
    (v, v) = [int(v) for v in input().split()]
    v = [int(v) for v in input().split()]
    f(((('STRING' + f(v)) + 'STRING') + f(f(v , v , v))))
```

Table 2: Generated edits (middle and right column) for given input programs (left column) in the test set. Strings, variables and function names are canonicalized.

these latent codes as targets. It has shown tremendous success in generative modeling for images, and largely avoids the posterior collapse issue that is prevalent in continuous VAEs (Kingma & Welling, 2013). We are therefore able to use VQ-VAEs with powerful Tranformer-based encoders and decoders, thus reaping the benefits of both large language models and discrete representations.

# 6 CONCLUSION

In this work, we apply discrete categorical transformations to source code. These transformations are aimed at increasing code efficiency, but could be useful for a variety of natural language tasks or discrete structures such as molecules or graphs.

We take one step towards learned models that improve code efficiency by demonstrating that discrete transformations can be learned from a supervised dataset – and that the VQ-VAE provides one architecture for doing so. To move beyond this paper and enable automatic optimization, canonicalization is the first challenge that needs to be solved. There are many potential solutions, like using specialized pointer networks for variables or using a Transformer pre-trained on a very large code corpus.

As we move towards building more ML models that can write code, it is critical that we also consider computational efficiency. Code that is run in datacenters across the world today contributes to not only cost, but also the carbon footprint of machines. We believe that optimizing this footprint is an exciting application of deep learning.

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

# A APPENDIX

## A.1 EDIT-VQVAE PSEUDOCODE

---

**Algorithm 1** Forward pass of Edit-VQVAE

---

**Require:** Input sequence $\mathbf{x}$, output sequence $\mathbf{y}$, encoder $f_\theta(\cdot)$, decoder $g_\phi(\cdot)$, embedding dictionary $Z$, positional encodings $PE$

1: $\mathbf{z}_{\text{fast}} \leftarrow f_\theta(\mathbf{x})$
2: $\mathbf{z}_{\text{slow}} \leftarrow f_\theta(\mathbf{z})$
3: $k \leftarrow \operatorname{argmin}_k \|\mathbf{z}_{\text{fast}} - \mathbf{z}_{\text{slow}}\|^2$
4: $\mathbf{z}_{\text{edit}} \leftarrow Z_k$
5: **return** $g_\phi(\mathbf{x} + PE + \mathbf{z}_{\text{edit}})$

---

---

**Algorithm 2** Generating from Edit-VQVAE

---

**Require:** Input sequence $\mathbf{x}$, output sequence $\mathbf{y}$, encoder $f_\theta(\cdot)$, decoder $g_\phi(\cdot)$, embedding dictionary $Z$, positional encodings $PE$

1: $\mathbf{z}_{\text{fast}} \leftarrow f_\theta(\mathbf{x})$
2: $\mathbf{z}_{\text{slow}} \leftarrow f_\theta(\mathbf{z})$
3: $k \leftarrow \operatorname{argmin}_k \|\mathbf{z}_{\text{fast}} - \mathbf{z}_{\text{slow}}\|^2$
4: $\mathbf{z}_{\text{edit}} \leftarrow Z_k$
5: **return** $g_\phi(\mathbf{x} + PE + \mathbf{z}_{\text{edit}})$

---

## A.2 ADDITIONAL FIGURES

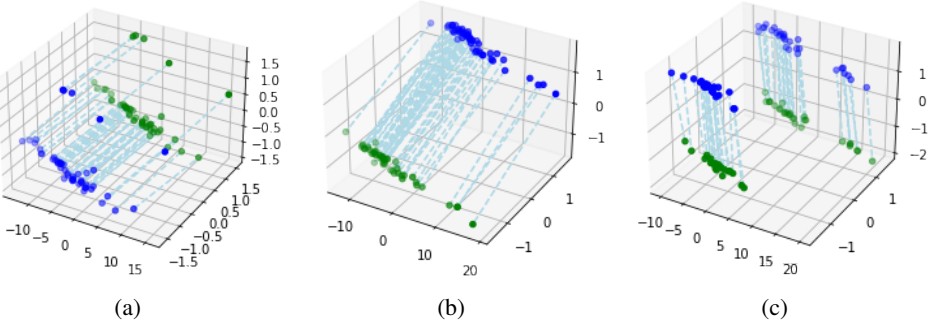

| (a) | (b) | (c) |

Figure 5: 3D PCA visualization of the program pairs assigned to the same learned latent variable. (a)/(b)/(c) are program pair samples from latent #8/#20/#40, respectively. The blue dots represent input (slow) programs. Upon making the edit, the programs move to become the green dots, the corresponding output (fast) programs. We observe that the fast programs belong to a different region of latent space than the slow programs and that within each latent category, the programs tend to move together - demonstrating that the edits that each latent applies tend to be coherent (or similar).

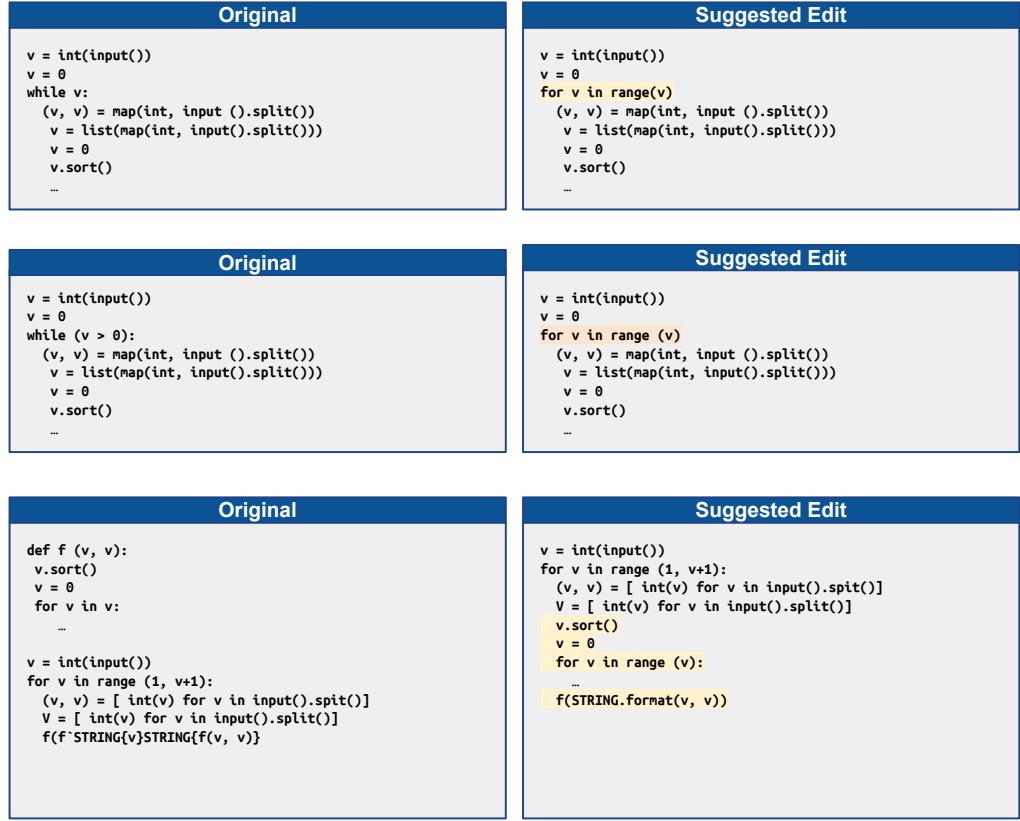

Figure 6: Sampling programs from latent #20 above, we see that the edits that the latent proposes tend to be similar. In the first two examples, a while loop is turned into a ranged for loop. In the third, a more complex edit is proposed. The function call is inlined into the original loop body and the for loop argument is changed to a range statement. For the other latents pictured in Figure 5, latent #8 proposes using a map data structure and latent #40 factors code into functions.

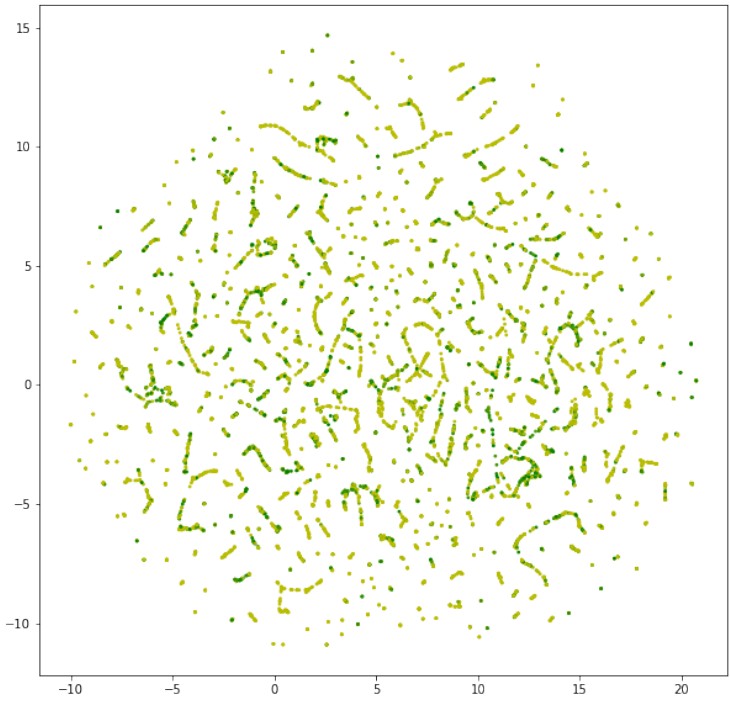

Figure 7: 2D UMAP visualization of the learned latent space. The programs are colored by runtime interpolated between 0.01 seconds (yellow) and 5 seconds (green). The yellow/green dots tend to cluster together.This color pattern exists locally, instead of globally.

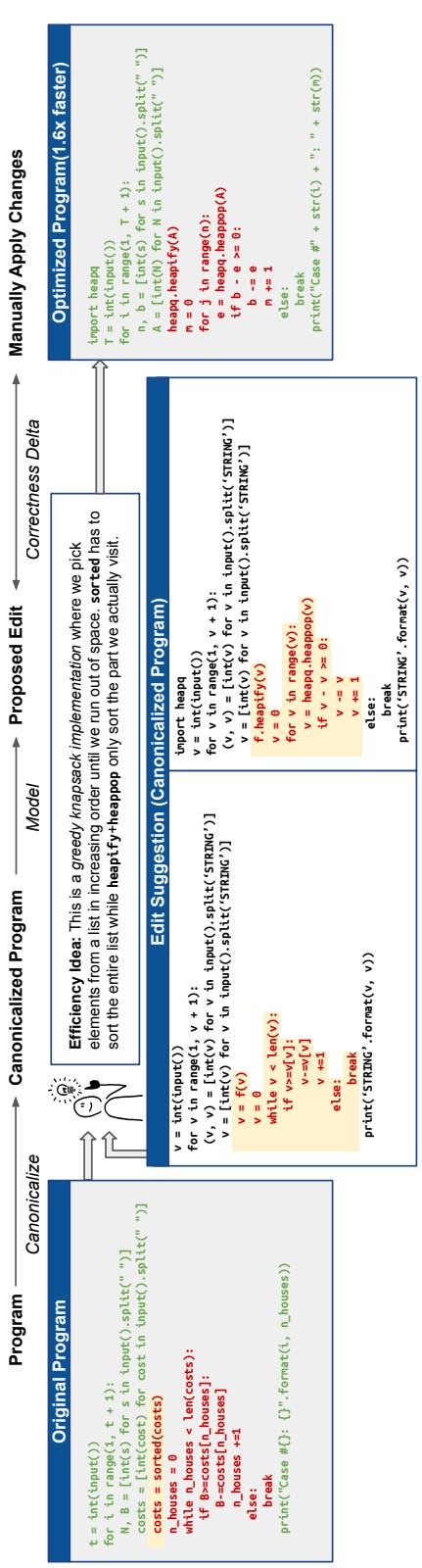

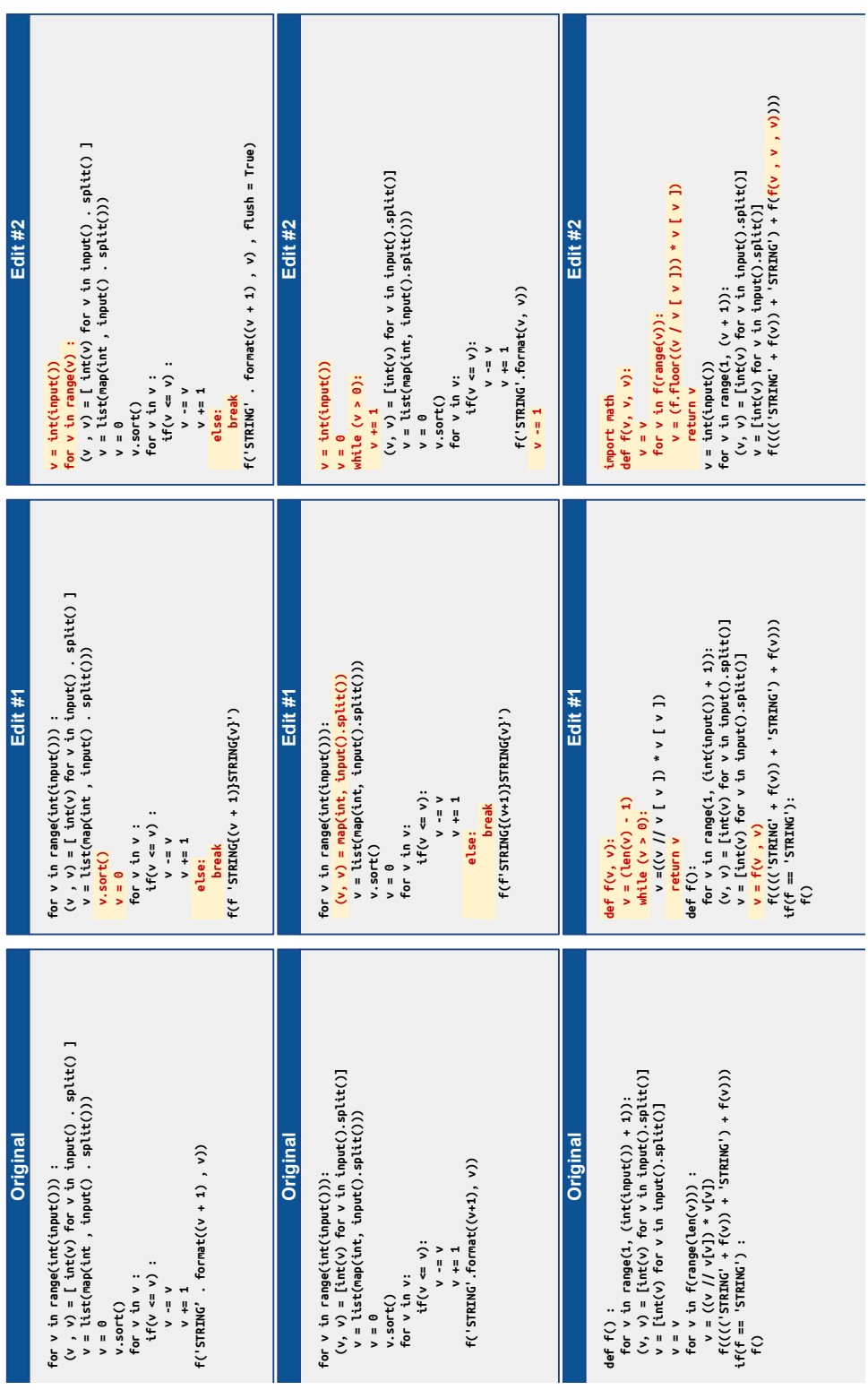

