# OpenReview forum: "Learning to Improve Code Efficiency"
_ICLR.cc/2023/Conference — Submitted to ICLR 2023_

### Official Review · Reviewer_39mZ · 2022-10-18

**Confidence:** 4
**Correctness:** 2
**Technical Novelty And Significance:** 4
**Empirical Novelty And Significance:** 2
**Recommendation:** 5

**Clarity, Quality, Novelty And Reproducibility:**

As a note on my background, I am not sufficiently familiar with the relevant education and software engineering literature to be able to judge the usefulness of the proposed model of suggesting code edits.

## Clarity

Overall, the paper is very clear. There are minor issues with the description of the approach, but these are not major concerns (and did not significantly impair my understanding of the paper).

* The ROUGE-8 metric is not described in the paper
* Figure 1, Figure 4: are these help texts ("Algorithmic Difference: Use a...", "Efficiency Idea: This is...", etc) generated by the approach? I didn't see any description of these, though during my initial reading I assumed they were. It would be more clear if the Figure 1 used phrasing that made it clear that these were descriptions, rather than prescriptions, and if Figure 4 did not include this text.
* Figure 3, and approach broadly:
  * The roles of the "prior sampler" and "commitment_loss" in Figure 3 are not discussed in the text
  * "The total input to the decoder is...". Does the decoder not also take the raw text of the slow program? Or is this only used during training?
* Nit: "Additionally, Figure 2(b) indicates that the data is multi-modal". Figure 2(b) only shows a single real mode of program lengths, around 125 tokens.

## Quality

The approach is a well-thought-out approach and seems like a reasonable solution to the task.
The quantitative aspect of the evaluation is also well done.
However, the qualitative aspect of the evaluation is lacking, and it's not clear to what extent the generated suggestions are practically useful, diverse, or interpretable.

There is essentially no qualitative analysis of the edits. What types of edits are learned? How interpretable are the different $z_i$, as the paper claims? The paper provides a couple of examples in Table 2, but this analysis isn't sufficient.
* E.g.: does the edit in Table 2(a) applied to other code blocks always result in adding early termination to a while loop? What happens if it is applied to a program without such an optimization opportunity? Do any other latent edits make this same change?
* In 2(b), Edit 2 seems to speed up the program by making the loop never execute (and doesn't apply the edit discussed in the caption)
* It's also not clear to what extent the results in Table 2 are randomly sampled v.s. cherry picked

The utility of these edits would ideally be demonstrated with a user study, to support the utility of the model and the underlying assumption that directionally (but not necessarily literally) correct edits are a useful hint for programmers.
Barring that, the paper should provide a more detailed qualitative analysis (including potentially a detailed set of case studies, randomly sampled edits, a user study, analyses of how consistent and interpretable the $z_i$ are, or other systematic empirical analyses).

The quantitative analysis is well-done, but not sufficient for understanding the results -- all of these scores have clear issues in the case where the generated transformation is correct and useful, but does not already exist in the dataset.

## Novelty

The approach is novel to the best of my knowledge, in that it provides a new problem formulation for helping users write more efficient code.

--------

## Update after author response

Thanks to the authors for their response. Based on this response, my score remains unchanged.

My main concern with the paper is that the core design decision of the VQVAE model, and several of the claims around it, are essentially unvalidated. For instance, the contributions section claims:

> We qualitatively demonstrate that the learned discrete latent variables represent different edits, and that the edits that are assigned to one latent variable are generally consistent.

The paper does not give sufficient evidence to validate this claim.

* There are not enough methodological details to understand Table 2. Are these proposed edits from the same or different latents?
* Table 2 also does not give enough examples to convincingly argue that there is consistency or diversity among the proposed edits.
* In Figure 5, showing that the edits apply what looks like a translation in PCA to 3 dimensions does not convincingly show that these edits are semantically consistent.
* Figure 6 is a step in the right direction, with a more clear statement of methodology showing that the edits applied are from the same latent. However, the data presented here is still very sparse, only showing some amount of consistency for one latent across three similar programs. The data do not show that different latents generate different edits.
* I'm also generally concerned about whether these results are cherry picked. The paper doesn't include any statement of methodology for how these examples are selected.


Having looked more at these results, I'm also concerned about the quality of the data, and the corresponding semantics of the task. All of the original programs in Figure 6 (and many of the other programs throughout the paper -- e.g., those in Table 2) either immediately crash or don't execute anything (but would crash if they were to execute). What does it mean to apply an edit to speed these programs up?


**Strength And Weaknesses:**


Strengths:
* Interesting problem under study, interesting and novel solution to the problem
* Very clearly described approach
* Strong quantitative evaluation of the results

Weaknesses:
* Weak qualitative evaluation of the results (see "Quality" below)


**Summary Of The Paper:**


The paper presents an approach to suggesting code changes that improve the performance of programs.
The paper first analyzes the distribution of (correct) solutions to competitive programming problems, finding a large gap between the efficiency of median-performance and top-quantile-performance solutions.
The paper proposes a generative approach to suggesting these code changes, based on a discrete VAE style approach.
To train, the approach uses Transformer encoders to encode a program and a corresponding optimized version of that program, takes the difference between the embeddings to generate an edit vector, discretizes the edit vector, and then decodes a new program using that discretized edit vector.
At evaluation time, the approach selects a random edit vector from a dictionary of trained edit vectors (representing different possible program edits), then decodes that using the original slow program to produce a new program with that edit applied.
The authors find that this approach qualitatively results in programs with optimizations applied, and quantitatively results in programs becoming more similar to semantically equivalent, optimized programs.


**Summary Of The Review:**

Weak reject.
Though the paper shows promise, the near complete lack of a qualitative evaluation means that the paper does not rise to the level of acceptance.
I would be willing to raise the score if the authors provide any of the following:
* A user study demonstrating that the suggested edits lead to better code performance (while maintaining correctness)
* A significantly more thorough analysis of the different $z_i$, showing the extent to which they consistently lead to the same proposed types of edits (on the same original programs and across different programs), the extent to which they encode semantically different types of edits with different $z_i$,
* Generally, more detailed case studies of proposed edits

---

> ### Author Response · Authors · 2022-11-19
> **Response to Reviewer 39mZ**
>
> We directly address the questions posed.
>
> “Clarity issues”
> We have noticed the minor clarity issues the reviewer mentioned and are working on improving the writing of the paper by incorporating the suggestions. Particularly, in Figure 3,
> - The prior sampler and commitment_loss are part of the VQVAE.
> - Both z_edit and the slow program will be fed to the decoder. In training time, z_edit is sampled from the posterior (it comes from the two tower architecture). In inference time, z_edit is sampled from the prior (uniform over the dictionary).
>
> We will make them clear in the next revision.
>
> “Qualitative studies”
> Please refer to Figures 5 and 6 in the new revision for the program pairs sampled from the same latent.

---

### Official Review · Reviewer_hCSu · 2022-10-19

**Confidence:** 5
**Correctness:** 1
**Technical Novelty And Significance:** 3
**Empirical Novelty And Significance:** 1
**Recommendation:** 3

**Clarity, Quality, Novelty And Reproducibility:**

Overall, the paper is clear and reproducible.
However, some figures such as Figure 1, Figure 4 and Table 2 contain tiny text, which makes them very difficult to read.

**Strength And Weaknesses:**

## Strengths
* The task of improving code efficiency is important and exciting
* The application of improving code efficiency is novel, as far as I know.
* The idea of curating a dataset of code competition solutions is clever, since these solutions can be executed and their runtime can be measured, and this runtime can serve as a signal for training.
* The proposed VQVAE model is interesting and performs better than the straightforward transformer (although not a contribution of this paper).

## Weaknesses
* Evaluation - The main weakness is evaluation. To train the model, the authors canonicalize the solutions in a way that does not allow to execute the edited programs anymore. That is, there is no way to measure whether the generated "improved" programs actually run faster than their original versions.
Instead, the authors use varieties of BLEU to measure whether the prediction is lexically similar to the reference. This measure is problematic, since lexical similarity does not guarantee faster execution. In fact, the solutions were clustered using lexical similarity initially (ROUGE-8), so the "slow" program and the "fast" reference program are already supposed to be quite lexically similar.

There is no need to generate executable programs in all papers; however in this paper, since it focuses on improving the efficiency of programs, demonstrating that the generated programs are indeed more efficient is crucial.

* Confusing terminology - some claims in the introduction are very confusing, and might not necessarily be correct.
Concretely, I think that the authors confuse "long runtime" with "low performance", and assume that every solution that takes a lot of time to run must be low-performance. For example, the introduction says `we find wide variance in computational cost: the runtime difference between a median solution and the 90th percentile solution is over two-fold`. But why are solutions to different problems comparable? This variance might stem from simply the variance in the questions, instead of the variance in the solutions, as the authors imply.

For example, an extremely inefficient solution for "binary search over 100 numbers", implemented in `O(n)` instead of `O(log(n))`, might run much faster than a super-efficient solution for "Bellman-Ford algorithm over a clique of 1000 nodes", but the inefficient solution might still run faster than the efficient solution, if we compare their absolute runtimes.

The paper concludes that `The scarcity of high-performance solutions highlight the difficulty of our task`, and `high-performance solutions are uncommon`. However, I was not convinced that high-performance solutions are scarce or uncommon. A solution may be high-performance - *but still have a long runtime*, because the problem it is solving requires polynomial time complexity, or its test inputs are longer, for example. In other words,
**a solution to a problem that requires polynomial time complexity is not necessarily inefficient** compared to the other potential implementations of the same problem.

* Analysis -  Since the VQ-VAE model is not the straightforward transformer, and the authors claim that `each discrete latent variable represents a different learned category of code-edit`, it would be nice to see some analysis of the learned latent variables and what have they captured, to understand why does the VQ-VAE perform so much better than the transformer.

## Additional Questions
1. The introduction says that `compilers and current tooling have more difficulty identifying higher-level optimizations, such as more efficient algorithms to the same problem`. Does the proposed model actually find "more efficient algorithms to the same problem"?
2. Why are Figures 2(a), 2(b) and 2(c) meaningful? Why is it meaningful to compare solutions of different problems? The variance in program length or runtime might be natural and stem from the variance in the questions. For example, one question might have logarithmic complexity and another question might have exponential complexity. Why should we compare their solutions' runtime, and conclude that the exponential one is "not high-performance"?
3. Does clustering of the submissions according to ROUGE-8 **really** find solutions to the same problem? Did the authors perform any manual annotation to verify that this is indeed the case? I can imagine examples where inter-problem solution similarity might be higher than intra-problem solutions. That is, solutions to different problems might be almost identical to each other, while different solutions to the same problem will be very different.
4. Section 4.3 say that `We use **syntactic** program similarity as a tool to measure model performance`. But in fact, all the metrics that are described below are variations of BLEU. Why is this *syntactic* similarity?


**Summary Of The Paper:**

The paper addresses the task of improving the efficiency of code, by providing feedback hints. The authors curated a dataset from the Google Code Jam competition, and trained a VQ-VAE model to automatically provide feedback on how to improve the runtime efficiency of the solutions.

**Summary Of The Review:**

The paper addresses an important problem using a clever source of data.
However, it suffers from a major evaluation weakness, since the main claim of "improving efficiency" could not be measured.
Thus, I must vote for rejection at this time, but I hope to see the paper's evaluation improved in the future.

I **will increase my score** if the evaluation could show an actual convincing improvement in runtime.

---

> ### Author Response · Authors · 2022-11-19
> **Response to Reviewer hCSu**
>
> We directly address the questions posed.
>
> “Main concern on evaluation”
> Please refer to the general response.
>
> “Confusing terminology”
> The Code Jam dataset we used runs across a diverse set of inputs, and we implicitly assume that these inputs are representative of what we care about, i.e., they are an inherent part of the problem definition. We use the largest/most complex testing dataset for each question during the evaluation. Longer runtime on this dataset constitutes a lower-performance code solution than a solution to the same question with a shorter runtime. We will make this more explicit in the next revision.
>
> “why are solutions to different problems comparable”
> They are not. When we say, “the runtime difference between a median solution and the 90th percentile solution is over two-fold”, we are comparing programs that answer the same question, on the same test-data. We will make this more explicit in the next revision.
>
> “A solution may be high-performance - but still have a long runtime, because the problem it is solving requires polynomial time complexity, or its test inputs are longer, for example. In other words, a solution to a problem that requires polynomial time complexity is not necessarily inefficient compared to the other potential implementations of the same problem.”
> “Does the proposed model actually find "more efficient algorithms to the same problem"?”
>
> The comparison of the execution time is within the same question, using the same set of test cases. In addition, in this paper, we only consider the execution time. That is to say, even if program A has a better time complexity than program B, we would still say A is more efficient than B if A has a much smaller execution time than B measured by the same set of inputs.
>
> “Analysis of the learned latents”
> Please refer to Figures 5 and 6 in the new revision for the program pairs sampled from the same latent.
>
> “Why are Figures 2(a), 2(b) and 2(c) meaningful?”
> Figure 2 presents the overall statistics of the dataset. In particular, 2(b) presents the long-tail distribution of program lengths. 2(c) presents the relative execution time between the programs in the same pair (from the same question), suggesting that there’s room for improvement for many existing programs. We agree that the runtime of different questions is not comparable and whenever we discuss relative runtime differences in the paper (in Figure 2(c) and elsewhere), it refers to pairs of solutions to the same problem, executed on the same input.
>
> “Does clustering of the submissions according to ROUGE-8 really find solutions to the same problem?”
> We restrict that the program pairs must come from the same question. We will make it more explicit in the next revision.

---

> > ### Comment · Reviewer_hCSu · 2022-11-19
> > **Thank you**
> >
> > Thank you for your response, and thank you for clarifying some of the points.
> >
> > >We agree that the runtime of different questions is not comparable
> >
> > For this reason, I believe that Figures 2(a) and 2(b) are completely meaningless.
> >
> >
> > The authors have not responded to all my questions.
> > I think that the paper will be a nice paper eventually, but currently, its evaluation is its main weakness.
> > The authors' responses about "having too few training examples" and "pretrained LLMs are not the focus" are not very convincing, as pretrained models of code are available and widely used.
> >
> > As many of the reviewers have mentioned, I think that the paper could be improved significantly by including a manual analysis to somewhat compromise for the weakness of the lack of execution-based evaluation. Such an analysis can demonstrate many of the claims in the paper, such as finding better algorithms, suggesting helpful hints, and showing the interpretability of these hints.

---

### Official Review · Reviewer_SDKK · 2022-10-24

**Confidence:** 3
**Correctness:** 3
**Technical Novelty And Significance:** 2
**Empirical Novelty And Significance:** 2
**Recommendation:** 5

**Clarity, Quality, Novelty And Reproducibility:**

The paper is well written.
In my opinion, there is some novelty in the VQ-VAE approach for improving efficiency of programs that authors propose.
The paper should be reproducible using the available dataset and description of methods in the paper.

------------------------------------------
ICLR review form does not provide a section for comments/questions about the paper. I am going to post them here.
- "We find that this canonicalization approach enables the model to learn on this dataset, but also means that we lose the ability to execute the edited program to evaluate run-time" - this rather compromises the correctness and efficiency evaluation. Are there ways to convert hints/changes back into the precannonicalized program form?
- "To quantify efficiency, we rely on textual similarity to other known correct code in the data-set that is more efficient." - for exact match, this should work well although it still has a problem of cannonicalized hints; for inexact match via dBLUE, this might miss performance regressions.
- It would be great if you tried using LMMs with a hint input with or without adaptation.


**Strength And Weaknesses:**

Strengths:

- Improved hint/change generation results by using proposed VQ-VAE instead of Transformer model
- In experiments the VQ-VAE performs better on hard efficiency metric that may indicate transformations that are correct and more efficient
- Ablation study of Edit VQ-VAE vs VQ-VAE and Edit-VAE

Weaknesses:

- Does not compare proposed VQ-VAE to results achievable from large language models (LLMs)
- Due to canonicalization of the code, authors cannot check if provided hints generate a correct and efficient program. Instead they use similarity metric to determine if program with changes/hints is correct and more efficient. This shortcut may be problematic if changes/hints only seem to be correct and more efficient, but in practice are not. This needs to be solved for more accurate evaluation in the future.

**Summary Of The Paper:**

Authors propose a variational auto encoder that suggests code edits to improve performance of programs. The system is trained using programs from a Google Code Jam competition. Data also contains the performance of the solutions. Authors show improvement of their VAE model vs sequence-to-sequence baseline model.

**Summary Of The Review:**

I think that the paper introduces interesting and well performing approach using VQ-VAE to provide hints/changes to improve efficiency of programs.
I think that some additional work like comparing the approach to using LMMs would improve this paper.
Ideally, it would be interesting if hints/changes could be translated back to pre-cannonicalized program form and programs evaluated there. Alternatively it would be good if authors did a user study and evaluation of providing the hints/changes.

---

> ### Author Response · Authors · 2022-11-19
> **Response to Reviewer SDKK**
>
> We directly address the questions posed.
>
> “More accurate evaluation and user study and evaluation of providing the hints/changes.”
> Please refer to the general response.
>
> “Does not compare proposed VQ-VAE to results achievable from large language models (LLMs)”
> In the paper, we already compared the proposed method to a similarly sized, and computationally comparable Transformer baseline, which demonstrated that the VQ-VAE we introduced in this work helped improve the model’s ability to discover the multi-modal code edits toward the more efficient program space.  We think that these experiments are representative of using an LLM as the base model. That said, we agree that using a pre-trained code LLM as the base model would help to avoid using the variable normalization trick. However, this is not the focus of this work.

---

### Author Response · Authors · 2022-11-19
**Common Concerns about Evaluations Metrics**

We thank all the reviewers for their comments, suggestions, and questions. Here we will first address the common question on evaluation metrics. More specific questions will be addressed in individual responses.

All reviewers expressed concerns about the text similarity-based evaluation metrics. Both reviewers SDKK and hCSu worried about the inconsistency between the similarity metrics and program correctness and efficiency. Reviewer 39mZ worried about the coverage of the reference fast programs.

While measuring the correctness and efficiency of the generated programs by directly executing them would be ideal, we have ~90000 unique python programs in the training dataset. Empirically, our experiments demonstrated that this number could not meet the sample requirements for generating programs with unnormalized variable names. We believe that this issue could be resolved by either using a much larger training program set, e.g., the training set for Github Copilot, or finetuning a pre-trained large language model trained on programs. Although we agree that generating programs with long-tail distributed unnormalized variable names is an important research problem, we believe that such a problem is orthogonal to the goal of this paper, which is to propose an architecture that learns interpretable, multi-modal code edits.

Similarly to measuring the quality of machine translation, we leveraged a performance-weighted text similarity metric to evaluate our generated programs. The reference programs in the test dataset provide many different ways that developers have implemented a solution to the same problem. Therefore, the performance-weighted text similarity with the reference programs can be used as a surrogate for the actual efficiency. We also include an exact match score to demonstrate the model’s ability to generate good programs with normalized variable names. Large language models such as Codex [1] have demonstrated that using more training data, variable names are learnable.

Taking a step back, we agree with reviewer SDKK that an approach that can reliably apply optimizations to programs, produce correct code that can be compiled, and allow us to compare the efficiency of the resulting program would be ideal. Realizing this vision is the overarching goal for this work, but it would require a number of unsolved problems in code learning to be solved. We don’t think that such an evaluation is achievable with the current state of the art.

Due to the magnitude of the end-to-end problem, we believe that it needs to be solved one portion at a time, and our evaluation methodology is reflective of this. We assume that the model would not be used end-to-end to generate an improved solution but would guide the programmer towards such a solution (i.e., a programmer would still need to make changes to the proposed program, similar to suggestions by other code models). We therefore think that our soft distance metric is appropriate – it is true that a proposed program could be incorrect, but a small distance means that a programmer would only need to make small amounts of changes to create a program that we know to be both correct and faster. This evaluation metric is therefore reflective of what the model is trying to achieve.

However, we do want to capture the “ideal” case as well. To this end, the hard efficiency metric captures how often the generated program is exactly correct and also faster. Under this metric, we can show actual improvements in runtime, as suggested by reviewer SDKK. Since we know that the data matches programs that are actually in our data set and that we have the runtime for, this does represent the “ideal” case from above – but only for a subset of programs.

We agree with 39mZ that as an early project in this area, qualitative understanding of the kinds of edits that the model learns and how a programmer may leverage them is particularly valuable. Figure 4 is an example of such a case study, where we show an actual end-to-end performance improvement and the edit that leads to it. We think that this is as close as possible to the “ideal” evaluation described above, even though only for one example. Due to our large library of problems and programs, we would be able to add more such case studies, possibly in an appendix due to space limitations.

[1] Chen, Mark, et al. "Evaluating large language models trained on code." arXiv preprint arXiv:2107.03374 (2021).

---

### Author Response · Authors · 2022-11-19
**New Paper Revision**

In the new revision, we added program pair examples in Figure 6 to show that the edit suggestions produced by the same latent are similar to each other. Specifically, in the first two examples, a while loop is turned into a ranged for-loop. In the third, a more complex edit is proposed. The function call is inlined into the original loop body, and the for-loop argument is changed to a range statement. For the other latents pictured in Figure 5, latent #8 proposes using a map data structure, and latent #40 factors code into functions.

---

### Decision · Program_Chairs · 2023-01-20

**Decision:**

Reject

**Justification For Why Not Higher Score:**

N/A

**Justification For Why Not Lower Score:**

N/A

**Metareview: Summary, Strengths And Weaknesses:**

This paper proposed a generative model trained on programming competition submissions to transform programs into faster versions of those programs. While the paper contains some interesting idea for an important problem, reviewers raised major weakness concerns about weak empirical evaluations, lack of convincing experimental results and strong baselines from the literature. While the authors tried to respond to the review questions,  some of major review concerns remain and the overall quality of this work is below the acceptance bar.